# Strain Engineering of Domain Coexistence in Epitaxial Lead-Titanite Thin Films

**Yanzhe Dong [1], Xiaoyan Lu [1,*], Jinhui Fan [2], Si-Young Choi [3] and Hui Li [1,*]**

1   School of Civil Engineering, Harbin Institute of Technology, Harbin 150001, China; dyz121025@163.com
2   Functional Materials and Acousto-Optic Instruments Institute, School of Instrumentation Science and Engineering, Harbin Institute of Technology, Harbin 150080, China; fan648895@163.com
3   Department of Materials Science & Engineering, Pohang University of Science and Technology, Pohang 37673, Korea; youngchoi@postech.ac.kr
*   Correspondence: luxy@hit.edu.cn (X.L.); lihui@hit.edu.cn (H.L.)

**Abstract:** Phase and domain structures in ferroelectric materials play a vital role in determining their dielectric and piezoelectric properties. Ferroelectric thin films with coexisting multiple domains or phases often have fascinating high sensitivity and ultrahigh physical properties. However, the control of the coexisting multiple domains is still challenging, thus necessitating the theoretical prediction. Here, we studied the phase coexistence and the domain morphology of $PbTiO_3$ epitaxial films by using a Landau–Devonshire phenomenological model and canonic statistical method. Results show that $PbTiO_3$ films can exist in multiple domain structures that can be diversified by the substrates with different misfit strains. Experimental results for $PbTiO_3$ epitaxial films on different substrates are in good accordance with the theoretical prediction, which shows an alternative way for further manipulation of the ferroelectric domain structures.

**Keywords:** ferroelectric epitaxial films; Landau–Devonshire method; domain coexistence; substrate misfit strain

## 1. Introduction

Phase and domain structures are crucial to dielectric and piezoelectric responses in ferroelectric materials [1–3] and, furthermore, the coexistence of multiple phases/domains exhibit unique structures and physical properties, opening a new window for sensitive mechanical sensors with higher dielectric and electromechanical responses [4,5]. To this end, tremendous efforts have been made to explore the ferroelectric phase transition and domain formation both theoretically and experimentally [6,7]. However, it has still been a big challenge to achieve precious control of domain structures in ferroelectric epitaxial thin films [8]. It is therefore necessary to understand the mechanism of the domain formation for further manipulation of the multiple domains and phases in ferroelectric thin films.

Such a coexistence of multiple phases/domains in ferroelectric thin films has been investigated by controlling misfit strains of substrate and growing conditions [4–6]. Lead-titanite ferroelectric thin films ($PbTiO_3$), usually, in the typical tetragonal phase can form $c$, $a/c$ and $a_1/a_2$ domain structures [4]. The domain structures become diverse in films with different substrates, where they can change from almost pure $c$ to a mixing state of $c$ and $a/c$, then a mixing $a/c$ and $a_1/a_2$ state; and finally, to $a_1/a_2$ state as the substrate misfit strain increases from compressive to tensile strains [9–12]. Specifically, Li et al. illustrated the thickness-dependence of the $a_1/a_2$ domain fraction of $PbTiO_3$ films grown on a single substrate [9]. Langenberg et al. reported the thickness-dependence and substrate strain-dependence of domain morphology in $PbTiO_3$ films and the effect of an applied electric field on domain structure distribution [10]. Nesterov et al. concluded the domain pattern in epitaxial $PbTiO_3$ films depends on the film thickness, miscut angle and growth speed [11]. Johann et al. found that interface type, substrate symmetry and miscut location direction

affects the domain structures as well [12]. Damodaran et al. described the effect of substrate strain on the degree of competition near the phase boundary between the $a/c$ and $a_1/a_2$ domain structures by experiments and phase field simulations [8]. Surprisingly, Lu et al. obtained epitaxial PbTiO$_3$ thin films that exhibit abnormal mechanical-force-induced large-area, non-local, collective ferroelastic domain switching near the critical tensile misfit strain [7]. This "domino-like" domain switching was attributed to the coexistence of $a/c$ and $a_1/a_2$ nanodomains with a small potential barrier in-between [13].

The energy potential between the multiple phases is extremely lowered due to their structural competition based on the phenomenological Landau–Devonshire theory [7,9–11]. In perovskite ferroelectrics with the multiple phases, the strain/stress state could be complex and monoclinic phases are usually formed under mechanical distortions [14]. The phase diagrams of perovskite ferroelectric films clarify that the multiple phases/domains are stabilized at the critical misfit strain [13,15,16]. These multiple domain structures near the phase boundary possess nearly degenerated energies and could coexist on a large scale [17] and the fraction is dependent on the difference of the coexisting energy potential, which has a great impact on the formed domain structures [18]. Recently, thermodynamic analysis and phase-field simulations further confirm that large piezoelectric and dielectric responses arise at morphotropic phase boundaries, which may be attributed to the dense domain wall [19].

As aforementioned, the multiple domains are crucial to the physical properties of ferroelectric materials especially near the critical points with coexisting phases/domains; however, it is still unclear about the phase/and domains distribution and formation of the final domain structures in epitaxial ferroelectric thin films. In this study, Landau's phenomenological theory was used to study the phase and domain coexistence in a PbTiO$_3$ epitaxial thin film [7]. The free energy of each coexisting domain state was compared with the consideration of misfit strain, domain size and temperature, which are vitally important for the domain formation. The volumetric fraction of coexisting domain structures was also calculated and compared with experimental results.

## 2. Materials and Methods

The PbTiO$_3$ has a high phase transition temperature (673 K), and a large spontaneous polarization of about $P = 0.7$ C/m$^2$. The films were epitaxially grown on single crystal substrates with nearly the same lattice parameters; therefore, we consider few defects were induced in the film growth. PbTiO$_3$ films are fabricated at 800–900 K by pulsed laser deposition. The domain structure begins to form at about 600 K [20]. Therefore, we investigate the domain coexistence both at the temperatures for domain initialization and finalization. To study the polydomain coexistence in ferroelectric thin films, Landau theory with the consideration of polydomain mechanical interaction was adopted [21]. We assume that the film is epitaxially uniform with a transversely isotropic misfit strain and the energy density is only composed of Landau free energy and elastic strain energy without the influence of depolarization fields. The renormalized thermodynamic potential after the Legendre transformation of the Gibbs free energy can be expressed with respect to the primary order parameters of polarization $P_i$ and internal mechanical stresses $\sigma_i$ in the film [9,21] as follows:

$$
\begin{aligned}
F ={} & a_1\left(P_1^2 + P_2^2 + P_3^2\right) + a_{11}\left(P_1^4 + P_2^4 + P_3^4\right) + a_{12}\left(P_1^2 P_2^2 + P_2^2 P_3^2 + P_3^2 P_1^2\right) + a_{123} P_1^2 P_2^2 P_3^2 \\
& + a_{111}\left(P_1^6 + P_2^6 + P_3^6\right) + a_{112}\left[P_1^2\left(P_2^4 + P_3^4\right) + P_2^2\left(P_3^4 + P_1^4\right) + P_3^2\left(P_1^4 + P_2^4\right)\right] \\
& + \tfrac{1}{2}s_{11}\left(\sigma_1^2 + \sigma_2^2 + \sigma_3^2\right) + s_{12}(\sigma_1\sigma_2 + \sigma_2\sigma_3 + \sigma_1\sigma_3) + \tfrac{1}{2}s_{44}\left(\sigma_4^2 + \sigma_5^2 + \sigma_6^2\right)
\end{aligned}
\tag{1}
$$

where $F$ is the total free energy, $a_i$, $a_{ij}$, $a_{ijk}$ are the linear and nonlinear dielectric stiffness coefficients, $s_{ij}$ and $\sigma_i$ are the elastic compliances and mechanical stresses, respectively. Parameters for our calculations are taken from Ref. [22]. According to experiments, there are three types of domain structures in PbTiO$_3$ films: pure $c$ domain, $a/c$ domain, and $a_1/a_2$ domain.

For the pure *c* domain structure, the expression of the total free energy and spontaneous polarization could be written as Equations (2) and (3) [23].

$$F_c = \frac{u_m^2}{s_{11} + s_{12}} + a_3^* P_c^2 + a_{33}^* P_c^4 + a_{111} P_c^6 \tag{2}$$

$$P_c^2 = \frac{-a_{33}^*}{3a_{111}} + \left( \frac{a_{33}^{*2}}{9a_{111}^2} - \frac{a_3^*}{3a_{111}} \right)^{1/2}, \tag{3}$$

where $a_3^* = a_1 - 2Q_{12}u_m/(s_{11} + s_{12})$ and $a_{33}^* = a_{11} + Q_{12}^2/(s_{11} + s_{12})$ with $u_m$ the substrate misfit strain and $Q_{ij}$ the electrostrictive coefficents. Following the polydomain theory, the spontaneous polarization and the corresponding total energies for *a/c* domain structures could be calculated by using the following Equations (4) and (5) [21].

$$F_{ac} = \frac{u_m^2}{2s_{11}} + a_3^* P_{ac}^2 + a_{33}^* P_{ac}^4 + a_{111} P_{ac}^6 \tag{4}$$

$$P_{ac}^2 = \frac{-a_{33}^*}{3a_{111}} + \left( \frac{a_{33}^{*2}}{9a_{111}^2} - \frac{a_3^*}{3a_{111}} \right)^{1/2}, \tag{5}$$

where $a_3^* = a_1 - Q_{12}u_m/s_{11}$ and $a_{33}^* = a_{11} + Q_{12}^2/2s_{11}$. Similarly, the analytical expression of the spontaneous polarization and the corresponding free energies for $a_1/a_2$ domains are described by Equations (6) and (7) [23].

$$F_{aa} = \frac{u_m^2}{s_{11} + s_{12}} + a_1^* P_{aa}^2 + a_{11}^* P_{aa}^4 + a_{111} P_{aa}^6 \tag{6}$$

$$P_{aa}^2 = \frac{-a_{11}^*}{3a_{111}} + \left( \frac{a_{11}^{*2}}{9a_{111}^2} - \frac{a_1^*}{3a_{111}} \right)^{1/2}, \tag{7}$$

where $a_1^* = a_1 - (Q_{11} + Q_{12})u_m/(s_{11} + s_{12})$ and $a_{11}^* = a_{11} + (Q_{11} + Q_{12})^2/[4(s_{11} + s_{12})]$.

According to the canonic statistical method, the volume fraction of domain structure in PbTiO$_3$ films could be calculated by thermodynamic probability. For each statistically equivalent ensembles *i*, the distribution probability of the ensemble being in the energy level $G_i$ can be written as Equation (8).

$$\gamma_i \propto \exp\left( -\frac{G_i - G_0}{kT} \right), \tag{8}$$

where *k* and *T* are the Boltzmann's constant and temperature, respectively. $G_i = F_i V_i$ is the energy for the *i*th different temperature domain structure with $F_i$ the system free energy densities and $V_i$ is the corresponding domain structure volume. $G_0$ is the energy for the ground state. The existing volume fractions for each phase are $f_i = \gamma_i/(\gamma_c + \gamma_{ac} + \gamma_{aa})$ with $I = c$, *ac*, *aa* that refer to the possible existing *c* domain, *a/c* domain, and $a_1/a_2$ domain structures.

Since ferroelectric domain structures in PbTiO$_3$ films initially form at around 600 K, it is necessary to investigate the domain distribution at high temperatures prior to the final state at room temperature [20]. We note that the intrinsic exiting properties were studied without consideration of the interaction between multiple domain structures.

## 3. Results and Discussion

The total free energy of PbTiO$_3$ films with respect to misfit strains at 600 and 300 K are shown in Figure 1a,b. The gray, orange and blue curves for pure *c* domain, *c/a* domain and $a_1/a_2$ domain in Figure 1a,b were calculated by Equations (2)–(7), respectively. The domain fractions at 300 and 600 K in Figure 1c,d were calculated by Equation (8). Due

to the great interest of recent research on ferroelectric films under tensile strain, we here focus on the formation of the multiple domains near the critical tensile strains, where the total free energies of $a_1/a_2$ domain and $a/c$ domain are equal at critical strains of $u_m = 0.46\%$ at 300 K and $u_m = 0.26\%$ at 600 K. As shown in Figure 1c,d, the volume fraction of $a/c$ domain sharply decreases with the increase of tensile strain, leading to $a_1/a_2$ domain dominating state at a higher tensile strain. The coexisting area shrinks with the decrease of temperature due to the larger energy difference of coexisting domain structures at lower temperatures. Besides, pure $c$ domain also coexists with a relatively small volume fraction under small tensile strain. It is worthwhile noting that the existing volume fraction changes with a temperature. For instant, $a_1/a_2$ domain dominates in the film with a misfit strain of $u_m = 0.46\%$ at high temperatures, while $a_1/a_2$ and $a/c$ co-dominate at room temperature. Therefore, the initial domain structure could be frozen with slightly higher elastic energy until further external stimuli were involved. It is also the reason for large-area, non-local, collective ferroelastic domain switching reported in PbTiO$_3$ epitaxial thin films [7].

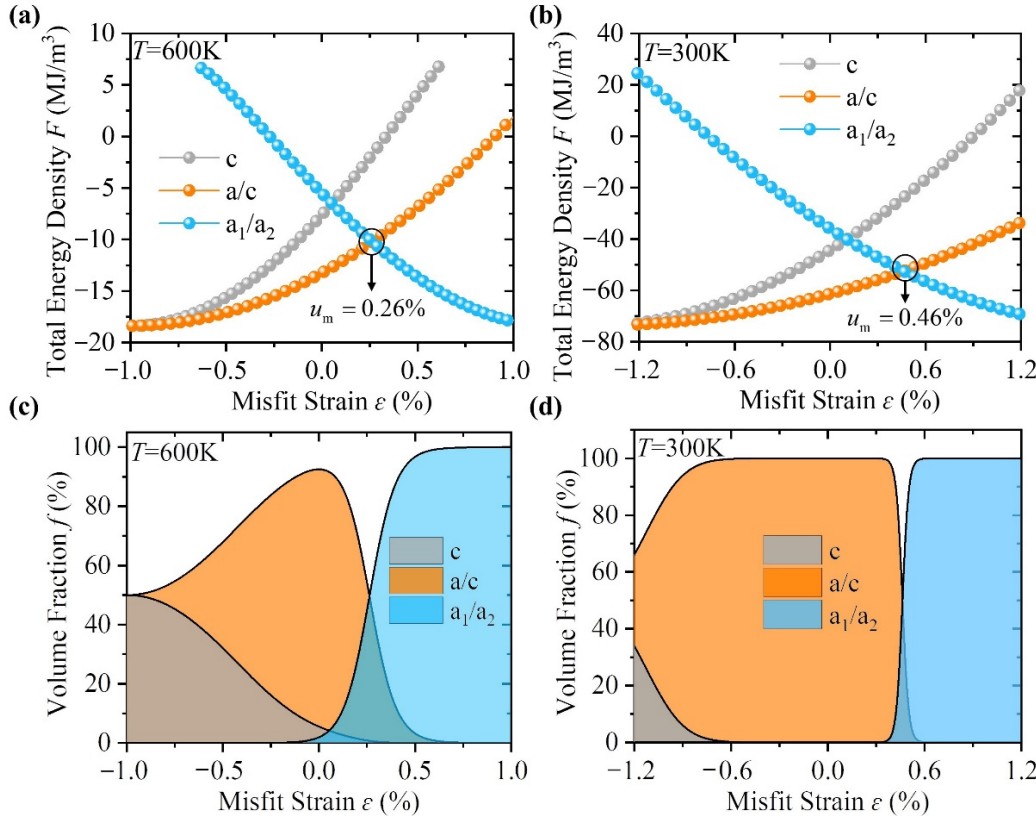

**Figure 1.** Total free energy of each possible domain structure in epitaxial PbTiO$_3$ thin with substrate strain variation at temperatures of (**a**) 600 K and (**b**) 300 K. (**c,d**) are the corresponding domain fractions. Overlapped areas show the domain coexistence.

To further investigate the domain coexistence and domain evolution during the film growth, we calculate the spontaneous polarizations of the $a_1/a_2$ and $a/c$ domains at the two typical temperatures. Without loss of generality, the domain size was considered as the same for each domain structure. The polarization-dependent total free energies in Figure 2a,c were calculated by using Equations (4) and (6) at different misfit strain and temperature. The temperature-dependent polarizations under different misfit strains were calculated by Equations (5) and (7). Inserts were calculated by using Equation (8). As shown in Figure 2a, the PbTiO$_3$ film consists of coexisting $a_1/a_2$ domain and $a/c$ domain under the critical strain of 0.26% at 600 K (Red curves), while the $a_1/a_2$ domain dominates at 300 K due to the lower energy (Blue curves). Since the spontaneous polarization decreases with the increase of temperature as shown in Figure 2b, it is likely that these two domain

structures more easily coexist at higher temperatures. With the decrease of temperature, initially formed domains slightly grow up with considerable increase of elastic energy, which could also limit the transformation of the initial domain structure to the more stable domain structure at low temperatures. Such initial-heritage domain structure will be kept during the cooling process and becomes relatively unstable due to the emergence of more stable states.

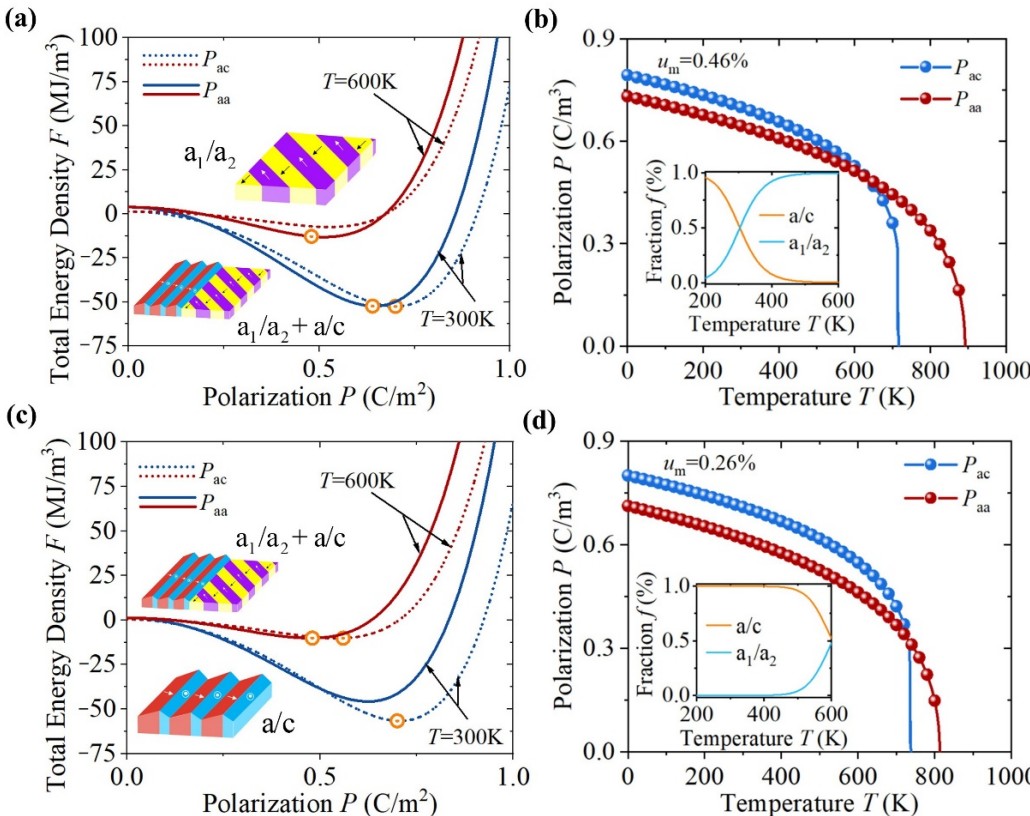

**Figure 2.** Calculated total free energy of $a_1/a_2$ domain and $a/c$ domain at 300 and 600 K with strains of (**a**) 0.46% and (**c**) 0.26%. Temperature-dependent polarization under misfit strain of (**b**) 0.46% and (**d**) 0.26%. Inserts are corresponding domain fractions.

Misfit strain at room temperature is an important factor for film design. For PbTiO$_3$ thin film, the critical substrate tensile strain is 0.46% at 300 K with $a/c$ and $a_1/a_2$ domain coexistence as shown in Figure 2c. It is reasonable that spontaneous polarizations with similar values are easier to switch in-between. Therefore, films under a tensile strain of 0.46% will be easier to have coexisting domains compared with that under strain of 0.26% as shown in Figure 2b,d. We note that the spontaneous polarizations for these two typical domain structures are the same at about 650 K with a substrate strain of 0.46%, which could be a critical temperature for aggressive domain competition and domain size shrinkage due to the easy switching between existing domain structures.

It is reported that coexisting domain structures commonly exist in epitaxial lead-titanite thin films with a misfit strain ranging from −0.1% to 0.6% [9–12]. In order to investigate the domain coexistence, we first compare the free energy density of each possible domain under various misfit strains as shown in Figure 3. The energy difference between existing domain structures is quite small at 600 K. Therefore, it is likely that each domain structure is aggressively competing to exist. Domain size could be determined by the competing intensity. We could speculate that the domain size of the initially formed domain structures is related to the free energy of coexisting domains, and could be smaller if the competition is more aggressive.

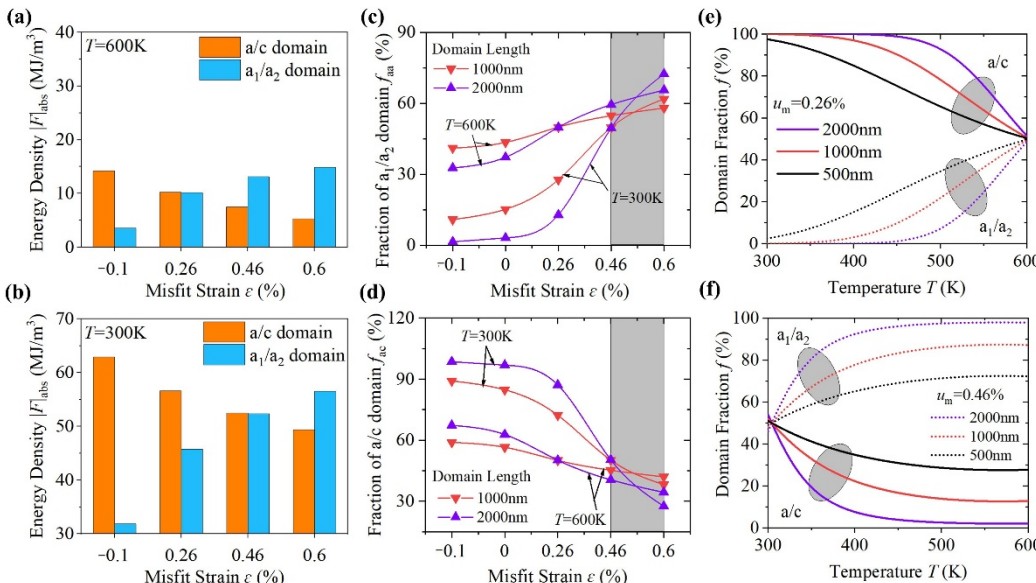

**Figure 3.** Total energy density difference between $a_1/a_2$ domain and $a/c$ domain under misfit strains of $-0.1\%$, $0.26\%$, $0.46\%$, $0.6\%$ compared at (**a**) 600 K and (**b**) 300 K. (**c,d**) Misfit-strain dependent domain fractions of $a_1/a_2$ and $a/c$ domains at 600 K and 300 K. Temperature-dependent domain fraction with different domain size under critical strains of (**e**) 0.26% and (**f**) 0.46%.

To quantitively examine the domain competition, we theoretically compare the energy difference between $a_1/a_2$ and $a/c$ domains under various misfit strains at 600 K and 300 K as shown in Figure 3a,b, respectively. The energy difference between $a_1/a_2$ and $a/c$ domains is lower than 10 MJ/m$^3$ for film with tensile strains at both temperatures, while for films with compressive strain, the energy difference at 300 K is much higher than that at 600 K, indicating the multiple domains preferentially exist in films under the tensile strain. To examine the corresponding existing volume fraction, each domain volume is also a key factor. Although the domain size is dependent on each specific system, we here fix the domain thickness of 70 nm and the domain width of 40 nm to examine the effect of the domain length on the domain existing fraction. As shown in Figure 3c,d, the $a_1/a_2$ and $a/c$ domains more easily coexist at higher temperatures with a nearly equal total volume fraction. It is interesting to note that the domain fractions change little with temperature under misfit strain ranging from 0.46% to 0.6% (grey area in Figure 3c,d), which could be helpful for the construction of frustrated systems since the competition of the existing domains can be kept during the cooling process. The existing fraction changes sharply with the domain size as shown in Figure 3e,f. For a domain with a length of 500 nm and the above-mentioned thickness and width, the coexistence of $a/c$ and $a_1/a_2$ domains can be kept in a larger range of temperature in the film under a misfit strain of 0.46% than that of 0.26% during the cooling process.

To verify the theoretical prediction, we compare the results with experiments for (001)-oriented epitaxial PbTiO$_3$ thin film grown by pulsed-laser deposition on substrates with different misfit strains. Epitaxial PbTiO$_3$ films were grown at 670 °C in a dynamic oxygen pressure of 50 mTorr at a laser repetition rate of 10 Hz, and a laser fluence of 1.9 J/cm$^2$ with the same condition as reported in Ref. [7]. The film thickness is 70 nm with an epitaxial Ba$_{0.5}$Sr$_{0.5}$RuO$_3$ layer as bottom electrodes. We choose (001)$_C$-SrTiO$_3$, (110)$_O$-GdScO$_3$, (110)$_O$-SmScO$_3$, and (110)$_O$-NdScO$_3$ substrates with nominal misfit strains of about $-0.1\%$, 0.3%, 0.46% and 0.7%, respectively (where the *O* denotes orthorhombic indices). The domain structure is studied by piezoresponse force microscopy (PFM) carried out on an MFP-3D (Asylum Research, Berkeley, CA, USA) using Ir/Pt-coated conductive tips (Nanosensor, PPP-NCLPt, NanoSensors, Neuchatel, Switzerland). The PFM imaging scan uses alternating current (AC) driving voltage of 1 V in dual AC resonance tracking

(DART) mode. Here, we only use topographic images of the films to identify the domain fractions of the well-known tetragonal-based domain structures.

As shown in Figure 4a–d, topographic images show distinct domain structures in films on different substrates. In film on a $(001)_C$-SrTiO$_3$ substrate causing $-0.1\%$ compressive strain, $a/c$ domains dominate together with only about 5% $a_1/a_2$ domains. With an increase of tensile strain, the $a/c$ domain decreases and $a_1/a_2$ domains become dominating on the stage, which is in good accordance with theoretical predictions as shown in Figure 4e–h. For film on $(110)_O$-SmScO$_3$, the fractions of $a_1/a_2$ and $a/c$ domains are nearly equal, indicating the more aggressive competitions of existing domains that can be described by the MPB (multiple phase boundary). Despite the slight mismatch between the theoretical prediction and experimental result under 0.46%-tensile strain, which may be due to more complicated domain formation conditions in this system [11,20], our calculations can explain the role of multiple domains as a function of temperature, strain, and domain size.

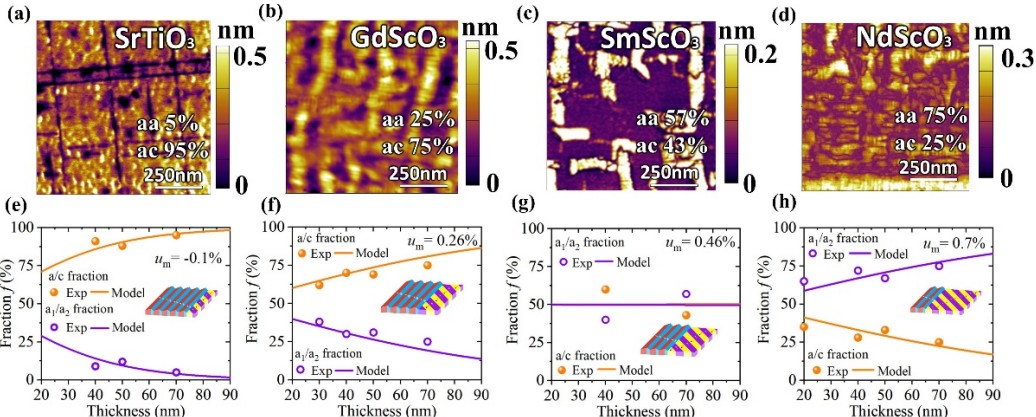

**Figure 4.** Topographic images of 70-nm-thick PbTiO$_3$ films with substrate strains of (**a**) $-0.1\%$, (**b**) 0.3%, (**c**) 0.46% and (**d**) 0.7%. (**e**–**h**) Theoretical calculation of thickness dependence of $a_1/a_2$ and $a/c$ domain fraction with different misfit strains at 300 K. Our experimental data and results from Refs. [8–10] were noted by orange circles for $a/c$ domains and purple circles for $a_1/a_2$ domains.

## 4. Conclusions

In summary, we studied the domain distribution in ferroelectric lead-titanite thin films using the Landau–Devonshire phenomenological model with the canonic statistical method. The coexistence and existing fraction of $a/c$ and $a_1/a_2$ domains in PbTiO$_3$ films were analyzed with the consideration of epitaxial misfit strain, temperature and domain size. Our results are in good accordance with experimental results for PbTiO$_3$ thin films grown on various substrates. This new methodology opens a new window for manipulating ferroelectric domain structures with ultrahigh-sensitivity and multi-state memories.

**Author Contributions:** Data curation, Y.D.; Investigation, Y.D.; Methodology, X.L.; Project administration, X.L. and H.L.; Supervision, X.L.; Writing—original draft, Y.D., X.L. and J.F.; Writing—review & editing, S.-Y.C. All authors have read and agreed to the published version of the manuscript.

**Funding:** This research is supported by the National Key Research and Development Program of China (No. 2021YFF0501001), Touyan Team Program of Heilongjiang Province, China, the National Science Foundation of China (No. 11872019), and the Fundamental Research Funds for the Central Universities (LH2020A006). S.-Y.C. acknowledges the support of a Korea Basic Science Institute (National Research Facilities and Equipment Center) grant funded by the Ministry of Education (2020R1A6C101A202).

**Institutional Review Board Statement:** Not applicable.

**Informed Consent Statement:** Not applicable.

**Data Availability Statement:** The data presented in this study are available upon request from the corresponding author.

**Conflicts of Interest:** The authors declare no conflict of interest.

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
