# Peer review of "Strain Engineering of Domain Coexistence in Epitaxial Lead-Titanite Thin Films"

_coatings, doi:10.3390/coatings12040542_

Round 1

Reviewer 1 Report

The paper is devoted for domain coexistence investigations in epitaxial lead-titanate thin films.

The topic is generally interesting, however the paper contains unexplained places (below) and need major revisions.

Please explain why in Eq. 1 are included members with higher power of P, like P^6?

Please clearly indicate according to which equations were calculated curves in Figs. 1-2.

Please explain meaning of points in Fig. 2.

In Fig. 4, has the meaninig the comaparison of one experimental point with theoretical curve?

Maybe more experimental data can be taken from literature for comparison?

Measurements units and number should be written separately, for example 300 K, not "300K".

Please explain why films of thickness 70 nm were investigated.

Reviewer 2 Report

This is a quite worthy article, which should undoubtedly be recommended for publication, but only after some improvement.

  1. It is not clear what has been done in this area in recent years, there is no literature review.
  2. More information about investigated PbTiO3 films is needed.  Especially, it is not clear what their defectiveness is.
  3. Note that construction of a model of polar nanoregions in the PMN relaxor ferroelectric based onfirst-principles lattice dynamics for chemically ordered supercells [S.A. Prosandeev et al., Phys. Rev. B 70,134110 (2004)], combined with invariance under permutations and dipole-dipole interaction as a source supporting randomly oriented residual polarization has been reported by Klotinsh https://doi.org/10.2478/s11534-010-0144-3 
  4. The role of the surface and especially its defectiveness especially needs to be quantified.

Round 2

Reviewer 1 Report

Authors make proper corrections according to reviewer remarks

and I suggest to publish the paper as it is.

Reviewer 2 Report

the authors have significantly improved the manuscript, and it can be recommended for publication